# Single-Ion Magnetism of the [Dy^III^(hfac)_4_]^−^ Anions in the Crystalline Semiconductor {TSeT_1.5_}^●+^[Dy^III^(hfac)_4_]^−^ Containing Weakly Dimerized Stacks of Tetraselenatetracene

**DOI:** 10.3390/ijms25158068

**Published:** 2024-07-24

**Authors:** Alexandra M. Flakina, Dmitry I. Nazarov, Maxim A. Faraonov, Ilya A. Yakushev, Alexey V. Kuzmin, Salavat S. Khasanov, Vladimir N. Zverev, Akihiro Otsuka, Hideki Yamochi, Hiroshi Kitagawa, Dmitri V. Konarev

**Affiliations:** 1Federal Research Center of Problems of Chemical Physics and Medicinal Chemistry RAS, Chernogolovka 142432, Russiamaksimfaraonov@yandex.ru (M.A.F.); 2Kurnakov Institute of General and Inorganic Chemistry, Russian Academy of Sciences, Moscow 119991, Russia; 3Institute of Solid State Physics RAS, Chernogolovka 142432, Russia; alexey_kuzmin_91@mail.ru (A.V.K.); khasanov@issp.ac.ru (S.S.K.);; 4Division of Chemistry, Graduate School of Science, Kyoto University, Sakyo-ku, Kyoto 606-8502, Japan

**Keywords:** tetraselenatetracene, semiconductor, single-ion magnet, crystal structure, magnetic coupling, optical and magnetic properties

## Abstract

The oxidation of tetraselenatetracene (TSeT) by tetracyanoquinodimethane in the presence of dysprosium(III) tris(hexafluoroacetylacetonate), Dy^III^(hfac)_3_, produces black crystals of {TSeT_1.5_}^●+^[Dy^III^(hfac)_4_]^−^ (**1**) salt, which combines conducting and magnetic sublattices. It contains one-dimensional stacks composed of partially oxidized TSeT molecules (formal averaged charge is +2/3). Dimers and monomers can be outlined within these stacks with charge and spin density redistribution. The spin triplet state of the dimers is populated above 128 K with an estimated singlet-triplet energy gap of 542 K, whereas spins localized on the monomers show paramagnetic behavior. A semiconducting behavior is observed for **1** with the activation energy of 91 meV (measured by the four-probe technique for an oriented single crystal). The Dy^III^ ions coordinate four hfac^−^ anions in [Dy^III^(hfac)_4_]^−^, providing *D*_2d_ symmetry. Slow magnetic relaxation is observed for Dy^III^ under an applied static magnetic field of 1000 Oe, and **1** is a single-ion magnet (SIM) with spin reversal barrier *U*_eff_ = 40.2 K and magnetic hysteresis at 2 K. Contributions from Dy^III^ and TSeT^●+^ paramagnetic species are seen in EPR. The Dy^III^ ion rarely manifests EPR signals, but such signal is observed in **1**. It appears due to narrowing below 30 K and has *g*_4_ = 6.1871 and *g*_5_ = 2.1778 at 5.4 K.

## 1. Introduction

Developing multifunctional materials is a very important task of modern science [1,2]. Up-to-date conducting materials are developed based on oxidized or reduced organic or organometallic molecules. They can show semiconducting properties, metallic conductivity, and even superconductivity, which is observed in some cases [3,4,5]. The field of molecular magnets is also growing quickly starting from compounds showing three-dimensional magnetic ordering of spins to single-ion magnets (SIMs), which have been successfully developed over the past three decades [6,7,8,9,10,11]. In contrast to classical magnets, SIMs can exhibit a slow magnetic relaxation of magnetization below blocking temperatures [12]. It is expected that such magnets are promising to be used in high-density memory devices [13]. Among different SIMs, lanthanide-based SIMs are most promising to reach high spin-reversible barriers and longer relaxation times. To date, the maximum blocking temperature was obtained for dysprosium SIMs containing substituted cyclopentadienyl ligands [8,9,10,14].

The combination of conductivity and SIM properties is also a very promising task since, potentially, conductivity and magnetism can coexist in one compound affecting each other. Previously, such phenomenon, known as giant magnetoresistance, was observed in molecular conductors when a magnetic field affects conductivity [15]. Such phenomena are interesting for spintronic applications. To date, several attempts have been made to combine such properties. For example, oxidized bis(ethylenedithio)tetrathiafulvalene (BEDT-TTF) or bis(ethylenedioxy)tetrathiafulvalene (BEDO-TTF) and M(dmit)_2_, where dmit is 4,5-dimercapto-1,3-dithiole-2-dithione, were co-crystallized with SIMs containing Re^IV^, Mn^II/III^, and Co^II^ ions [16,17,18,19,20,21,22,23]. In several cases, these compounds show a metallic type of conductivity in some ranges and SIM properties at very low temperatures. Lanthanide-based SIMs are also promising to be combined with oxidized or reduced organic molecules, which can manifest conductivity. Previously, we combined BEDT-TTF with different Ho, Dy, and Tb chlorides, but no SIM properties were found for these semiconductors [24]. Electrocrystallization was used to combine Dy^III^ magnetic chains and BEDT-TTF to prepare semiconducting SIMs [25]. A similar semiconducting SIM was obtained with 7,7,8,8-tetracyano-*p*-quinodimethane (TCNQ) coordinated with Dy^III^ [26]. Semiconducting SIMs were obtained based on BEDT-TTF and tetramethyltetrathiafulvalene (TMTSF) in the form of (BEDT-TTT)_5_Dy(NCS)_7_(KCl)_0.5_ and (TMTSF)_5_[Dy(NCS)_4_(NO_3_)_2_]CHCl_3_ crystals [27,28]. An interesting example of conductive SIM was obtained by Katoh et al. [29] when the double-decker Dy^III^Pc_2_ (Pc = phthalocyanine) complex was partially oxidized to [Dy^III^Pc_2_](I)_x_. It showed a metallic type of conductivity at room temperature, but it was semiconducting at lower temperatures (10^−3^ S·cm^−1^) probably due to 1D instability. The SIM had *U*_eff_ = 58 cm^−1^. It was conducting due to the partial oxidation of the Pc ligand, and the small negative magnetoresistance can be due to 4f−π interactions [29].

In this work, we developed a new synthetic approach for the compounds of oxidized organic molecules with metal hexafluoroacetylacetonate anions. Compounds based on Dy^III^(hfac)_3_ can potentially show SIM properties, for example, Dy^III^(hfac)_3_· (1,10-phenantroline-5,6-dione) is a SIM with *U*_eff_ = 194.3 K [30]. Therefore, we tried to obtain a compound of oxidized tetraselenatetracene with the [Dy^III^(hfac)_4_]^−^ anions. This compound was isolated as good-quality single crystals with the {TSeT_1.5_}^●+^[Dy^III^(hfac)_4_]^−^ (**1**) composition. This allows us to study its crystal structure, conductivity on single crystals as well as optical and magnetic properties. The TSeT stacks are not uniform, displaying a semiconducting behavior that is supported by band structure calculations. Semiconducting stacks coexist in **1** with the [Dy^III^(hfac)_4_]^−^ anions, which show SIM properties at liquid helium temperatures. It is interesting that TSeT^●+^ radical cations, according to our data, are weakly involved in magnetic coupling between Dy^III^ ions.

## 2. Results and Discussion

A new chemical method was developed to obtain salt {TSeT_1.5_}^●+^[Dy^III^(hfac)_4_]^−^ (**1**). One equivalent of tetracyanoquinodimethane (TCNQ), anhydrous Dy^III^(hfac)_3_, and a slight excess of TSeT (sublimed) were mixed in dry *o*-dichlorobenzene in anaerobic conditions for 24 h at 60 °C, providing the dissolution of TSeT and the formation of a deep green-blue solution and some amount of light-green powder precipitated. TCNQ oxidizes TSeT to a radical cation state, forming TSeT^●+^ and TCNQ^●−^ ions. The light-green powder contains mainly Dy^III^ and TCNQ^●−^ according to the microprobe analysis and IR spectra. Therefore, the insoluble Dy^III^(TCNQ)_3_ polymer can precipitate from the solution. This polymer can be similar to Dy^III^(TCNE)_3_, where TCNE is tetracyanoethylene studied previously by Miller et al. [31]. Therefore, Dy^III^(hfac)_3_ is not stable in the presence of TCNQ^●−^, and hfac^−^ ligands are substituted by TCNQ^●−^ forming an insoluble polymer. As a result, free hfac^−^ anions formed in the solution add to Dy^III^(hfac)_3_ forming [Dy^III^(hfac)_4_]^−^ anions that become counter anions for TSeT^●+^. The large size of [Dy^III^(hfac)_4_]^−^ provides high enough solubility of the salt despite the very poor solubility of TSeT^●+^ salts in organic solvents. The Dy^III^(TCNQ)_3_ polymer was filtered, and TCNQ was not found in the reaction products. Therefore, the resulting reaction solution contains only TSeT^●+^, [Dy^III^(hfac)_4_]^−^, and probably some amount of hfac^−^. That allows one to obtain salt **1** in a pure state as large well-shaped single crystals when the obtained *o*-dichlorobenzene solution is slowly mixed with *n*-hexane for two months. The preliminary reaction scheme (Figure 1) is shown below:

Though the reaction mechanism is not studied, only crystals of **1** were isolated without any powdered or crystalline admixtures. Neutral TSeT can also be present in the synthesis since a slight excess of TSeT is used relative to other components, allowing the formation of the salt with a partial +2/3 charge averaged on the TSeT molecules.

It should be noted that only some of metal hexafluoroacetylacetonates (La^III^(hfac)_3_ or M^II^(hfac)_2_) can be used in these syntheses, and salts obtained with TSeT and BEDT-TTF will be published in our future article. Other substituted metal acetylacetonates (like acac or 2,2,6,6-tetramethyl-3,5-heptanedionate (TMHD)) are stable in the presence of TCNQ^●−^, and no crystals were obtained in this case.

Optical properties of **1** are discussed in Appendix A.

The crystal structure of {TSeT_1.5_}^●+^[Dy^III^(hfac)_4_]^−^ (**1**) salt studied at 100.0(2) K contains one and a half independent TSeT and one independent Dy^III^(hfac)_4_ unit (Appendix A). Several CF_3_ groups of hfac are disordered between two orientations even at 100.0(2) K (only groups in major orientation are shown in the Figure 1a and Appendix A). The observed composition indicates a formal +2/3 charge averaged on the TSeT molecules since Dy^III^(hfac)_4_ units have an exact −1 charge due to an addition of one hfac^−^ anion to Dy^III^(hfac)_3_. The [Dy^III^(hfac)_4_]^−^ anions form three-dimensional packing due to the presence of shortened van der Waals C…F, O…F, and F…F contacts. Some of these contacts are rather short, for example, several F…F contacts are in the 2.81–2.90 Å range. π-stacks composed of the TSeT cations are formed in this three-dimensional network (Figure 1a). The geometry of the [Dy^III^(hfac)_4_]^−^ anions is shown in Appendix A. Totally, four hfac^−^ anions coordinate to each Dy^III^ and, as a result, it is surrounded by eight oxygen atoms. We analyzed the geometry of the surrounding of Dy^III^ with the SHAPE 2.1. software [32,33] (Appendix A) and found that the smallest value is obtained for the triangle dodecahedron geometry with *D*_2d_ symmetry (0.374). However, the square antiprism geometry having *D*_4d_ symmetry is also positioned rather close to the minimum (2.855). Nevertheless, two sets of four oxygen atoms form two planes that are perpendicular to each other (the dihedral angle between them is 89.96°), and the deviation of oxygen atoms from these planes is small, being 0.023–0.032 and 0.031–0.041 Å. The length of the Dy-O bonds varies from 2.32 to 2.36 Å.

There are two independent TSeT molecules in the stacks. Molecules derived from one and half independent TSeT are marked as types A and B, respectively. They have slightly different lengths of the Se-Se bonds of the 2.3251(4) and 2.3308(4) Å for the molecules of type A and B, respectively. It is known that the transition from the neutral TSeT to TseT^●+^ is accompanied by changes in bond lengths [34,35,36]. The length of the Se–Se bond in the neutral TSeT is 2.335(3) Å [34]. The length of these bonds in the salts with a formal +0.5 charge on the molecule is 2.323(3) Å in TSeT_2_Cl, 2.320(7) Å in TSeT_2_(SCN), and 2.325(2) and 2.324(2) Å in TSeT_2_(Hg_2_I_6_) [34,35,36]. The length of the Se–Se bond for TSeT^●+^ with +1 charge on the molecule is 2.322(3) Å [34]. Thus, there is an obvious tendency to the shortening of the Se-Se bond when the positive charge on the molecule increases. Therefore, it is seen that the positive charge on TSeT of type B is smaller than that on TSeT of type A, providing a non-uniform charge distribution in the stacks. According to the length of the Se-Se bond, the charge on molecules of type A is between +0.5 and +1.0, and the charge on molecules of type B is between 0 and +0.5.

Molecules of type A form dimers with a parallel arrangement of TSeT (due to the inversion center between them) in an atom-over-atom manner with some shift relative to each other (Figure 1). The short interplanar distance of 3.441 Å provides short C…C, C…Se and Se…Se contacts, resulting in effective π-π stacking in the dimers. The TSeT molecule of type B is rotated relatively to the molecule of type A by 77.4°. The A-B interplanar distance is slightly shorter (3.423 Å), but the number of shortened van der Waals contacts is essentially less relative to the A-A dimer. Consequently, the π-π interaction is less effective in this case. Thus, the TSeT stacks can be considered to be composed of dimeric and monomeric units. 1D metallic conductivity is not expected in **1** because of the non-uniform charge distribution and the presence of two structurally different TSeT molecules. Several short F…Se contacts are formed for each TSeT with the surrounding [Dy^III^(hfac)_4_]^−^ anions of 3.28–3.34 Å length. The shortest Se-Dy distance is 7.14 Å for the molecules of type A and 6.78 Å for the molecules of type B.

The magnetic properties of **1** were studied by the EPR and SQUID techniques (Appendix A). The *χ*_M_*T* value is equal to 14.34 emu·K/mol at 300 K (Figure 2a). This value is only slightly higher than the theoretically calculated value of 14.17 emu·K/mol for Dy^III^ with the ^6^*H*_15/2_ electron configuration at *g* = 4/3. If we add the contribution from one *S* = 1/2 spin localized on TSeT, we obtain a close value of 14.33 emu·K/mol. It is seen that, from a magnetic point of view, the contribution from spins localized on TSeT is very small in comparison with the contribution from the Dy^III^ ion. The *χ*_M_*T* values are nearly temperature-independent down to 10 K, and the *χ*_M_*T* values decrease only below this temperature (Figure 2a). Such magnetic behavior can be described quite well by the PHI 3.1.5. software fitting for Dy^III^ [37], with *g* = 4/3 and zero-splitting parameter *D* = −5.2 cm^−1^. At such *D* value, the averaged intermolecular coupling *zJ* was estimated to be −0.004 cm^−1^ (Figure 2a). Both the *D* value and the averaged intermolecular coupling (*zJ*) provide a decrease in the *χ*_M_*T* values at low temperatures. The determined set of parameters better describes the magnetic behavior of **1** at low temperatures. Additionally, salt **1** manifests a slight increase in the *χ*_M_*T* values below 50 K, as shown in the inset in Figure 2a. However, PHI fitting cannot reproduce well this increase, and it is very small in comparison with the observed *χ*_M_*T* values. As we discuss below in the EPR section, this increase can be due to weak antiferromagnetic coupling between Dy^III^ spins through the spins localized on the TSeT monomers, which preserve spin even at low temperatures. As a result, pairs with a parallel arrangement of Dy^III^ spins can appear. Potentially, such weak coupling is possible since the TSeT monomers are positioned between the Dy^III^ ions with a Se-Dy^III^ distance of 6.78 Å. At the same time, this distance is rather long, and spins are only partially localized on the TSeT monomers. Magnetization is not saturated for **1** even at 50 kOe magnetic field at 2 K, and the value of magnetization achieved at this field is 7.50 *μ*_B_*N*_A_ (Appendix A).

The dynamic magnetic susceptibility measurements in an oscillating magnetic field of 3 Oe were carried out to study the dynamic magnetic behavior of **1** (Appendix A). The measurements in a zero external magnetic field (*H*_DC_ = 0 Oe) showed no significant signals at 2 K (Appendix A). It is well known that the effect of quantum tunneling of magnetization (QTM) can be suppressed by the application of an external static magnetic field. We performed the AC measurements at applied external DC fields up to 5000 Oe, allowing us to observe weak but discernible out-of-phase signals on the *χ*″(*ν*) dependence that indicates slow magnetic relaxation in **1** (Appendix A). The optimal value of the DC field was set as 1000 Oe. The frequency (0.1–100 Hz) dependences of the in-phase and out-of-phase components of AC magnetic susceptibility for **1** under applied static field of 1000 Oe in the 2.2–3.8 K temperature range and the fit of the experimental data by the generalized Debye model are shown in Appendix A in the 0.1–100 Hz range since no peaks were found in the 100–1500 Hz range.

The resulting dependence of ln(*τ*) on inverse temperature for the relaxation process is shown in Figure 2c. The best fit of the experimental data was achieved using Equation (1), which is a linear combination of the Orbach, Raman, and direct mechanisms.
(1)$$\tau^{-1}=\tau_{0}^{-1}\exp\left(-\frac{U_{eff}}{k_{B}T}\right)+CT^{n}+AH^{4}T$$

The obtained anisotropy barrier is *U_eff_* = 27.9 ± 4.4 cm^−1^ (40.2 ± 6.3 K) for **1**. The other fitting parameters are *τ*_0_ = 7.5 (±0.9) × 10^−7^ s, *C* = 0.0011 ± 0.0002 K^−*n*^·s^−1^, and *A* = (3.6 ± 1.0) × 10^−12^ Oe^−4^·K^−1^·s^−1^. The exponent of the Raman mechanism was fixed as *n* = 9 [38]. Thus, salt **1** can be classified as a single ion magnet induced by magnetic field of 1000 Oe. The estimated spin reversal barrier of 40.2 K is rather small. The magnetic hysteresis loop for **1** was measured at 2 K (Figure 2b). It has a single-butterfly shape characteristic of SIMs [39]. However, the lines merge when the field approaches zero within the ±900 Oe range (Figure 2b). Such behavior is characteristic of field-induced SIMs, which are magnets in an external magnetic field (specifically, a 1000 Oe static field at 2 K). The hysteresis loop is collapsed at 5 K. The splitting between the curves obtained in zero-field (ZF) and field-cooling (FC) conditions is also characteristic for magnets. Weak splitting appears below 15 K (Appendix A), but an essential difference is observed only below approximately 5 K when salt **1** transfers to a single-ion magnet state.

Conductivity was measured by the four-probe technique for a single crystal oriented by X-ray diffraction in such a way that it is measured along the conducting 1D stacks from the TSeT molecules (along the crystallographic *a* axis). The resistivity of the crystal is about 2000 Ω·cm at 300 K, and resistivity increases with the temperature decrease (Appendix A). The estimated activation energy for this semiconductor is 91 meV (the determination of this energy in the cooling process is in the 300–210 K range provided in the inset in Appendix A).

Since the contribution from TSeT on the background of high-spin Dy^III^ ions is not resolved, SQUID data cannot provide information about the magnetic behavior of the TSeT sublattice. This information can be obtained by using the EPR technique. Salt **1** at 290 K shows an intense EPR signal, which can be described well by two lines (Figure 3a). The main line is rather broad and has *g* = 2.0177 and the linewidth (Δ*H*) of 15.36 mT. Besides this, a narrower line is observed at a larger *g*-factor of 2.0322, and the linewidth of this line is 3.6 mT. Since the integral intensities of the narrower line are only 1% of that of the broader line, we can conclude that this line originates from impurities. The broader line noticeably narrows with the temperature decrease (Figure 3e) down to about 6 mT at 150 K, and the *g*-factor only slightly decreases with the temperature. The most interesting feature of this line is that its intensity decreases with the temperature, indicating a temperature-activated process (Figure 3f). A minimal intensity is observed at 128 K (Figure 3f), and at this temperature, the intensity of the main signal decreases more than three times (2700 at 128 K and 9800 at 290 K in arb. units). If we plot the natural logarithm of the integral intensity of the signal at a certain temperature deducting the integral intensity of a signal at 128 K multiplied by reciprocal temperature, we obtain a linear dependence that allows us to determine the gap for this process, which is equal to 542 K (Figure 3f). Since TSeT stacks contain monomers (type B) and dimers (type A), we can suppose that dimers can show singlet–triplet transitions and the determined gap corresponds to a singlet–triplet transition in the dimers. Dimers are nearly diamagnetic at 128 K, but spins appear on them above this temperature. It is interesting that the appearance of spins on the dimers above 128 K can provide the growth of conductivity above 150 K. The population of the triplet state of the dimers can increase the number of carriers in the TSeT stacks. The signal can be attributed to the TSeT monomers below 128 K, and it shows a different behavior since the intensity of this signal increases with the temperature decrease. The signal splits at 128 K and below. As a result, this asymmetric signal can be fitted by three lines (Appendix A). An example of this signal is shown at 80 K, which can be fitted by three lines with the following parameters: *g*_1_ = 2.0240 and Δ*H* = 3.81 mT, *g*_2_ = 2.0087 and Δ*H* = 2.13 mT, *g*_3_ = 2.0006 and Δ*H* = 1.22 mT (Appendix A). This signal shows nearly paramagnetic temperature dependence with a small Weiss temperature of only −0.8 K (Appendix A). Taking into account the obtained Curie–Weiss dependence for spins localized on the monomers, we can conclude that the integral intensity of this signal at 290 K is about 1300 arb. units, whereas the integral intensity of the EPR signal localized on the dimers at 290 K is 8500 arb. units. Thus, following the crystal structure data, spin density is mainly localized on the TSeT molecules of type A. At the same time, the TSeT molecules of type B are not neutral and share about 14% of spin density at 290 K (this contribution can decrease due to the further population of a triplet state in the dimers). The EPR signal from the TSeT monomers is strongly broadened below 20 K. As a result, the signal at 5.4 K is nearly three times broader than the signal at 20 K. Due to the broadening of the line, splitting is not well resolved below 20 K, and the signal can be described better by one Lorentzian line. Generally, the temperature decrease provides a narrowing of EPR signals, but an essential broadening of the EPR signal can be due to the participation of the TSeT monomer spin in the magnetic coupling of spins. Since paramagnetic species below 20 K can be the Dy^III^ spins, the TSeT monomers can mediate magnetic coupling between them providing the formation of the Dy^III^ pairs with the parallel orientation of spins, such orientation provides the growth of the *χ*_M_*T* values below 50 K, as shown in the inset in Figure 2a.

Generally, high-spin paramagnetic lanthanide ions are not manifested in the EPR spectra due to high zero-field splitting, which takes place for dysprosium ions and usually significantly exceeds the energy of a microwave quantum at X-band EPR [40,41]. However, a very broad signal appears for **1** below 30 K. It has a linewidth of more than 250 mT at 30 K and cannot be well resolved. Therefore, at higher temperatures, this signal is not observed due to very large broadness. However, this signal strongly narrows with the temperature decrease, and it is well resolved at 12 K with the following parameters: *g*_4_ = 6.0826 (Δ*H* = 54.2 mT) and *g*_5_ = 2.1124 (Δ*H* = 164.0 mT) at 12 K (Figure 3b). The temperature decrease narrows even more this signal and shifts it to larger *g*-factors: *g*_4_ = 6.1871 (Δ*H* = 49.8 mT) and *g*_5_ = 2.1778 (Δ*H* = 137.6 mT) at 5.4 K (Figure 3c). This signal can be unambiguously attributed to Dy^III^ since its integral intensity is more than one-hundred times higher than that of the EPR signal of TSeT^●+^ (this signal is also well resolved in the EPR spectra at 12 and 5.4 K due to narrowness, Figure 3c). Namely, such ratio of the intensities is expected from the contributions of TSeT and Dy^III^ to magnetic susceptibility of **1** at low temperatures. The spectrum of Dy^III^ at the lowest available temperature of 4.2 K disappears most probably due to broadening (Appendix A). This effect can be attributed to the transition of {Dy^III^(hfac)_4_}^−^ to the single-ion magnet state.

The central panel of Figure 4 presents a zone diagram for one-dimensional stacks from the TSeT cations along the direction coming through the symmetric points in k-space Γ = (0, 0, 0) and X = (0.5, 0, 0) of the first Brillouin zone and a plot of the density of states (DOS). According to the calculated band structure, the system is a semiconductor with the energy gap *E*_gap_ = 103.17 meV. A possible reason for the appearance of the forbidden energy gap is the partial weak dimerization of the TSeT monomers in the stacks. The width of the forbidden energy gap is in agreement with the conductivity measurements for a single crystal of {TSeT_1.5_}^●+^[Dy^III^(hfac)_4_]^−^ and experimentally determined activation energy *E*_a_ = 91 meV. The Fermi level with energy *E*_F_ = −4.3789 eV attaches the upper boundary of a valence band, whereas the widths of unoccupied (conduction) and valence bands are *W*_u_ = 323.02 meV and *W*_v_ = 432.64 meV. In the tight-binding method, the value of the bandwidth (*W*) of the stack is determined by transfer integrals (*t*) between the neighboring TSeT monomers. This allows the transfer integrals in the stacks to be estimated as *t*_u_ = *W*_u_/2 = 161.51 meV and *t*_v_ = *W*_v_/2 = 216.32 meV.

The right panel of Figure 4 shows the diagram of the molecular levels of the isolated {TSeT}_3_^2+^ trimer and the TSeT monomer in the left and right sides of energy axis, respectively, calculated using the same level of theory. For ease of comparison, all energy levels of trimer and dimer are shifted by the Fermi energy level of the stack (−4.3789 eV). The analysis of the energy spectrum of the trimer and band structure of the stacks allows us to conclude that conduction and valence bands are composed mainly of the LUMO and the HOMO of trimers with *E*_LUMO_ = −4.2146 eV and *E*_HOMO_ = −4.8883 eV, and a substantial contribution to these orbitals is introduced by the atomic orbitals of the selenium atoms.

## 3. Materials and Methods

### 3.1. Materials

Tetraselenatetracene (TSeT) purified by sublimation was used. Dy^III^(hfac)_3_ was obtained via the interaction of a water solution of Dy^III^Cl_3_·6H_2_O with three equivalents of (K^+^)(hfac^−^) (Strem), also dissolved in water. A white powder precipitated, which was filtered and dried under vacuum and heating for four hours at 80–95 °C (86% yield). The obtained powder was stored in a glove box. *o*-Dichlorobenzene (C_6_H_4_Cl_2_) was distilled over CaH_2_ under reduced pressure, and *n*-hexane was distilled over Na/benzophenone. The solvents were degassed and stored in a glove box. All manipulations for the synthesis of **1** were carried out in an MBraun 150B-G glove box in a controlled atmosphere and with contents of H_2_O and O_2_ of less than 1 ppm. The crystals were stored in the glove box and were sealed in 2 mm quartz tubes under the ambient pressure of argon for EPR and SQUID measurements. KBr pellets for the IR- and UV-visible-NIR measurements were prepared in the glove box.

### 3.2. Synthesis

TSeT (26 mg, 0.048 mmol), TCNQ (8.5 mg, 0.042 mmol), and Dy^III^(hfac)_3_ (31.7 mg, 0.042 mmol) were dissolved in 18 mL of *o*-dichlorobenzene during 1 day at 60 °C. The solution turned dark green-blue after stirring for one day. All components were dissolved, the solution was cooled down to room temperature and filtered into a glass tube of 46 mL volume, and *n*-hexane (26 mL) was layered over the obtained solution. Some amount of light-green powder remained on the filter. It was washed with *n*-hexane, and the microprobe analysis showed the presence of Dy^III^. The IR spectrum supported the presence of the CN vibrations of TCNQ^●−^ at 2100–2200 cm^−1^. The slow interdiffusion of *n*-hexane and *o*-dichlorobenzene for 2 months yielded good-quality black crystals with a nugget shape without any admixtures. The solvent was decanted from the crystals, and they were washed with *n*-hexane to obtain crystals at a 34% yield. The composition of crystals determined from X-ray diffraction was {TSeT_1.5_}[Dy^III^(hfac)_4_] (**1**). All crystals had the same shape and color. The same unit cell parameters for several crystals tested from the synthesis indicated that only one crystal phase was formed. The microprobe analysis showed the ratio of Dy to Se as 1.0 to 6.1, which is close to the determined composition.

Crystal data for **1** at 100.0(2) K: C_47_H_16_DyF_24_O_8_Se_6_, F.W. 1800.86, black nugget, 0.21 × 0.13 × 0.08 mm^3^, monoclinic, space group *P* 2_1_/n, *a* = 10.4467(3), *b* = 14.7447(5), *c* = 33.9861(11) Å, *β* = 92.1871(12)°, *V* = 5231.2(3) Å^3^, *Z* = 4, *d*_calcd_ = 2.287 M gm^−3^, *μ* = 5.743 mm^−1^, *F*(000) = 3392, 2*θ_max_* = 61.014°; 87,873 reflections collected, 15,933 independent; *R*_1_ = 0.0295 for 12,857 observed data [>2σ(*F*)] with 840 parameters and 261 restraints; *wR*_2_ = 0.0613 (all data); final GoF = 1.018. CCDC 2362822.

X-ray diffraction data for **1** were collected using a D8 Venture diffractometer (Bruker, Berlin, Germany) in the ϕ- and ω-scanning modes at the Center for Collective Use of the Kurnakov Institute of General and Inorganic Chemistry, the Russian Academy of Sciences (λ = 0.71073 Å, Incoatec IμS 3.0 microfocus X-ray source). Primary indexing, the refinement of unit cell parameters, and the integration of reflections were performed using the Bruker APEX3 (version 5.054) and SAINT (version 6.36A) software package [42]. Reflection intensity was corrected for absorption using the SADABS 2016/2 software. The structures were solved by direct method and refined by the full-matrix least-squares method against *F*^2^ using SHELX 2016 and Olex2 [43,44]. Non-hydrogen atoms were refined in the anisotropic approximation. Positions of hydrogen atoms were calculated geometrically. The crystal structure of **1** at 100.0(2) K contains several CF_3_ groups disordered between two orientations with the 0.826(11)/0.174(11), 0.653(15)/0.347(15), 0.883(5)/0.117(5), and 0.768(7)/0.232(7) occupancies.

### 3.3. General

UV-visible-NIR spectra were measured in KBr pellets on a Perkin Elmer Lambda 1050 spectrometer (Perkin Elmer, Shelton, CT, USA) in the 250–2500 nm range. FT-IR spectra were obtained using KBr pellets with a Perkin-Elmer Spectrum 400 spectrometer (400–7800 cm^−1^) (Perkin Elmer, Shelton, CT, USA). EPR spectra were recorded for a polycrystalline **1** from room temperature (RT) down to 4.2 K with a JEOL JES-TE 200 X-band ESR spectrometer equipped with a JEOL ES-CT470 cryostat (JEOL, Akishima, Tokyo, Japan). A Quantum Design MPMS-XL SQUID magnetometer (Quantum Design, San Diego, CA, USA) was used to measure the static magnetic susceptibility of **1** in the magnetic field of 1000 Oe under cooling and heating conditions in the 300–1.9 K range. The sample holder contribution and core temperature-independent diamagnetic susceptibility (*χ*_d_) were subtracted from the experimental values. The *χ*_d_ values were estimated from the extrapolation of the data in the high-temperature range and fitting the data with the following expression: *χ*_M_ = *C*/(*T* − *Θ*) + *χ*_d_, where *C* is the Curie constant and *Θ* is the Weiss temperature. The dynamic magnetic properties of **1** were studied at 3.0 Oe ac field at the 0.1–1500 Hz frequencies by a Quantum Design MPMS-5S SQUID magnetometer (Quantum Design, San Diego, CA, USA).

## 4. Conclusions

A new method for the preparation of metal hexafluoroacetylacetonate {M(hfac)_x_} salts with tetraselenatetracene was developed. This method uses the instability of some M(hfac)_x_ in the presence of TCNQ^●−^ that allows the precipitation of the insoluble M(TCNQ)_x_ polymer, but M(hfac)_x_ are dissolved as counter anions due to the addition of hfac^−^ to them. This approach yields a tetraselenatetracene salt with the [Dy^III^(hfac)_4_]^−^ anions. The salt contains one-dimensional alternating stacks of monomers and weakly dimerized TSeT molecules, the latter of which show a quasi-one-dimensional conductivity of the semiconducting type with the activation energy of 91 meV. This is caused by the nonuniform stacks consist of paramagnetic TSeT monomers and the TSeT dimers, which show a singlet–triplet transition with a population of a spin triplet state above 128 K. The population of the triplet state increases the number of carriers in the stacks, and the growth of conductivity is observed mainly above 150 K. Eight oxygen atoms coordinate to the Dy^III^ ion in [Dy^III^(hfac)_4_]^−^. Surrounding is closer to snub disphenoid or triangular dodecahedron, and such geometry can provide the appearance of SIM properties. For example, Dy^III^(hfac)_3_·(1,10-phenantroline-5,6-dione) with a similar geometry of the Dy^III^ surrounding also shows SIM properties [28]. SIM properties are also observed in [Dy^III^(hfac)_4_]^−^ with *U*_eff_ = 40.2 K. As a result, the compound shows hysteresis at 2 K. Since this SIM is induced by a magnetic field of 1000 Oe, a hysteresis loop collapses when the magnetic field approaches zero (±900 Oe). Magnetism and conductivity exist in **1** in different temperature ranges: SIM properties are observed below 5 K when the conductivity is suppressed, but the conductivity of the semiconducting type is manifested mainly above 128 K. Nevertheless, weak coupling between paramagnetic TSeT monomers and the Dy^III^ ions can produce random Dy^III^ pairs with a parallel arrangement of spins since the EPR signal from the TSeT monomers is strongly broadened at a low temperature, and a weak increase in the *χ*_M_*T* values is observed in **1** below 50 K. Dy^III^ ions unusually show EPR signal, which is more than one-hundred times more intense than the EPR signal from TSeT and is not observed above 30 K due to essential broadening. The EPR signal from Dy^III^ disappears at liquid helium temperatures due to the formation of a single-ion magnet state. The results of other crystalline salts of the M(hfac)_x_^−^ anions (M = metal) with the donors TSeT and BEDT-TTF will be published in our future article.

## Data Availability

Data are contained within the article or Appendix A.

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
