# Peer review of "Single-Ion Magnetism of the [DyIII(hfac)4]− Anions in the Crystalline Semiconductor {TSeT1.5}●+[DyIII(hfac)4]− Containing Weakly Dimerized Stacks of Tetraselenatetracene"

_ijms, 2024, doi:10.3390/ijms25158068_

Round 1
Reviewer 1 Report
Comments and Suggestions for Authors
The manuscript provides a new method for single-crystal synthesis of oxidized organic molecules with dysprosium hexafluoroacetylacetonate anions. This class of materials belongs to single-ion magnets (SIMs) with (semi)conducting behavior. The authors studied both magnetic and electric properties of this compounds. They found that the semiconducting property is given by the one-dimensional stacks of weakly dimerized TSeT molecules, while the magnetic susceptibility is mainly from the Dy ions, with non-negligible contribution from the TSeT stacks. The measured EPR signal clearly separates the contributions from Dy and the TSeT stacks. The manuscript is well-prepared and contributes significantly to the field of SIMs. The results and analysis were presented in a relatively clear manner. I only have a few minor suggestions (see below). After the authors address these, the paper can be considered for publication.
1. For Fig. 3, it would be better to use consistent color and symbol scheme between the upper and lower panels. In panels d and e, it would be better to show the fitting results for the Dy contributions (or add new panels for this purpose).
2. In line 177, does the author mean “antiferromagnetic” instead of “ferromagnetic”?
3. It would be better to add legends for different atoms in Fig. 1a.
Author Response
The manuscript provides a new method for single-crystal synthesis of oxidized organic molecules with dysprosium hexafluoroacetylacetonate anions. This class of materials belongs to single-ion magnets (SIMs) with (semi)conducting behavior. The authors studied both magnetic and electric properties of this compounds. They found that the semiconducting property is given by the one-dimensional stacks of weakly dimerized TSeT molecules, while the magnetic susceptibility is mainly from the Dy ions, with non-negligible contribution from the TSeT stacks. The measured EPR signal clearly separates the contributions from Dy and the TSeT stacks. The manuscript is well-prepared and contributes significantly to the field of SIMs. The results and analysis were presented in a relatively clear manner. I only have a few minor suggestions (see below). After the authors address these, the paper can be considered for publication.
- For Fig. 3, it would be better to use consistent color and symbol scheme between the upper and lower panels. In panels d and e, it would be better to show the fitting results for the Dy contributions (or add new panels for this purpose).
Response: For each line in the spectra we use the same color. The main signal from TSeT is shown by the blue line. The impurity signal at high temperatures is shown by the violet line. Components of split signal from TSeT by are shown by green and turquoise colors, and signal from Dy by orange and red lines.
Fitting for Dy is shown in Fig. 3c by red and orange lines and now one more spectrum at 12 K (Fig. 3b) we add from supporting information. Parameters of fitting for the EPR spectrum from Dy are given in the text. In our measurements signal from Dy is observed only at three temperatures and in this case any temperature dependencies for parameters of the EPR signal cannot be presented. A symbol scheme was added to the figure caption.
- In line 177, does the author mean “antiferromagnetic” instead of “ferromagnetic”?
Response: This sentence was corrected.
Additionally, salt 1 manifests a slight increase in the XMT values below 50 K as shown in the inset to Fig. 2a.
Interaction between TSeT and Dy is antiferromagnetic but such coupling can order spins of Dy parallel to each other increasing the XMT values.
- It would be better to add legends for different atoms in Fig. 1a.
Response: Legend for different atoms is added: “Carbon is dark brown; fluorine is green; dysprosium is violet; oxygen is red; selenium is light brown”.

Reviewer 2 Report
Comments and Suggestions for Authors
The present manuscript by A.M. Flakina et al entitled: “Semiconducting behavior of crystalline {TSeT1.5} +[DyIII(hfac)4] in weakly dimerized stacks of tetraselenatetracene and single ion magnetism of the [DyIII(hfac)4] anions.”reports on interesting material consisting of single-molecule magnet molecules bringing magnetic bistability and tetraselenatetracene molecules establishing conductive properties. As such, it brings enough of novelty needed for publication in IJMS after revisions.
Comments:
1) Page 3, Line 111: The authors mention future plans involving the preparation of a “full article” on a larger series of compounds. This should be rephrased, as the current manuscript is not a communication, which may confuse readers.
2) pg.3, L114: please verify if the measurement temperature was really 100(2) K or 100.0(2) K. Even if the authors did not describe the cooling setup used, standard devices such as Cryostream are capable of maintaining the desired temperature within ±0.2 K.
3) The SHAPE calculations should be checked for errors—this reviewer obtained significantly different results with much smaller Continuous Shape Measures values. The results are as follows:
Structure [ML8 ] OP-8 HPY-8 HBPY-8 CU-8 SAPR-8 TDD-8 JGBF-8 JETBPY-8 JBTPR-8 BTPR-8 JSD-8 TT-8 ETBPY-8
Ln , 33.440, 24.570, 12.706, 5.265, 2.855, 0.374, 15.915, 28.025, 3.572, 3.035, 3.789, 6.123, 25.473
4) The authors use the term “pi-pi stacking.” This is jargon; the interaction should be properly named (e.g., π-π stacking, aromatic ring interactions).
5) The authors fitted magnetic data, presumably using a standard spin Hamiltonian, which is not typically used for fitting data of 4f complexes with strong spin-orbit coupling. The authors should:
i) State the spin Hamiltonian used.
ii) Support its use with references to previously published works that employed it.
iii) Calculate isothermal magnetization at 2 K using the parameters resulting from the fit and compare the result with experimental data.
Comments on the Quality of English Language
The quality of English language is sufficient for publishing.
Author Response
The present manuscript by A.M. Flakina et al entitled: “Semiconducting behavior of crystalline {TSeT1.5} +[Dy(III)(hfac)4] in weakly dimerized stacks of tetraselenatetracene and single ion magnetism of the [Dy(III)(hfac)4] anions.”reports on interesting material consisting of single-molecule magnet molecules bringing magnetic bistability and tetraselenatetracene molecules establishing conductive properties. As such, it brings enough of novelty needed for publication in IJMS after revisions.
Comments:
- Page 3, Line 111: The authors mention future plans involving the preparation of a “full article” on a larger series of compounds. This should be rephrased, as the current manuscript is not a communication, which may confuse readers.
Response: This sentence was corrected and word “full” was deleted.
- 3, L114: please verify if the measurement temperature was really 100(2) K or 100.0(2) K. Even if the authors did not describe the cooling setup used, standard devices such as Cryostream are capable of maintaining the desired temperature within ±0.2 K.
Response: The real temperature is 100.0(2) K and we correct this in the manuscript.
3) The SHAPE calculations should be checked for errors—this reviewer obtained significantly different results with much smaller Continuous Shape Measures values. The results are as follows:
Structure [ML8 ] OP-8 HPY-8 HBPY-8 CU-8 SAPR-8 TDD-8 JGBF-8 JETBPY-8 JBTPR-8 BTPR-8 JSD-8 TT-8 ETBPY-8
Ln , 33.440, 24.570, 12.706, 5.265, 2.855, 0.374, 15.915, 28.025, 3.572, 3.035, 3.789, 6.123, 25.473
Response: New results are added to SI and discussed.
4) The authors use the term “pi-pi stacking.” This is jargon; the interaction should be properly named (e.g., π-π stacking, aromatic ring interactions).
Response: We correct this term to “π-π stacking”.
5) The authors fitted magnetic data, presumably using a standard spin Hamiltonian, which is not typically used for fitting data of 4f complexes with strong spin-orbit coupling. The authors should:
- i) State the spin Hamiltonian used.
- ii) Support its use with references to previously published works that employed it.
iii) Calculate isothermal magnetization at 2 K using the parameters resulting from the fit and compare the result with experimental data.
Response:
Fitting of magnetic data by PHI.
The spin Hamiltonian for the fitting of the susceptibility data of the complex 1 in PHI was used as
ZEE = µB⋅gJ ⋅ (1)
ZFS = D (2)
where S and B with hats, µB, gJ, D, and S refer to operators of spin, magnetic field, the Bohr magneton, g-factor for lanthanide, axial zero-field splitting (ZFS) constant, and total spin on the metal ion, respectively. The Dy(III) ion with ground state free ion term symbol 6H15/2 was considered. We added this information to the Supporting information. A similar spin Hamiltonian was used for simulating the magnetic data of Dy-containing organic conductor based on tetramethyltetraselenafulvalene [1].
Moreover, experimentally observed cMT value of 14.34 emu×K/mol at 300 K for complex 1 is close to the calculated value for free Dy (III) nucleus (14.17 emu×K/mol) [2] as well as Dy-containing complexes (13.66-14.25 emu×K/mol) [3,4]. A slight excess of the observed value can be due to additional contribution from paramagnetic TSeT. Good fitting of magnetization was not obtained but the description of the XMT vs. T curve is well. Magnetization theoretically calculated with the same parameters by PHI has higher value of magnetization then that is observed experimentally. We suppose some information with the fitting of magnetic data is useful for this manuscript.
- Wan, Q.; Wakizaka, M.; Zhang, H.; Shen, Y.; Funakoshi, N.; Che, C.-M.; Takaishi, S.; Yamashita, M. A New Organic Conductor of Tetramethyltetraselenafulvalene (TMTSF) with a Magnetic Dy(III) Complex. Magnetochemistry 2023, 9, 77, doi:10.3390/magnetochemistry9030077.
- Feltham, H.L.C.; Brooker, S. Review of Purely 4f and Mixed-Metal Nd-4f Single-Molecule Magnets Containing Only One Lanthanide Ion. Coordination Chemistry Reviews 2014, 276, 1–33, doi:10.1016/j.ccr.2014.05.011.
- Guo, F.-S.; He, M.; Huang, G.-Z.; Giblin, S.R.; Billington, D.; Heinemann, F.W.; Tong, M.-L.; Mansikkamäki, A.; Layfield, R.A. Discovery of a Dysprosium Metallocene Single-Molecule Magnet with Two High-Temperature Orbach Processes. Inorg. Chem. 2022, 61, 6017–6025, doi:10.1021/acs.inorgchem.1c03980.
- Gorshkov, E.V.; Korchagin, D.V.; Yureva, E.A.; Shilov, G.V.; Zhidkov, M.V.; Dmitriev, A.I.; Efimov, N.N.; Palii, A.V.; Aldoshin, S.M. Effect of Ligand Substitution on Zero-Field Slow Magnetic Relaxation in Mononuclear Dy(III) β-Diketonate Complexes with Phenanthroline-Based Ligands. Magnetochemistry 2022, 8, 151, doi:10.3390/magnetochemistry8110151.

Reviewer 3 Report
Comments and Suggestions for Authors
Preparing a weakly dimerized stack of tetraselenatetracene (TSeT) and single ion magnetism of the [Dy(III)(hfac)4]- anions combining conducting and magnetic sublattices, Flakina et al. studied the behavior of crystalline {TSeT1.5}.+[DyIII(hfac)4]- salt. The authors have thoroughly investigated the system, and the manuscript is written well. I have a few questions for the authors.
Two broad questions are:
(1) How does conducting behavior and magnetism of {TSeT1.5}.+[DyIII(fac)4]- compare with pure TSeT monomers and Dy?
(2) How localized is the magnetism resulting from Dy?
(3) The magnetic moment of a DyIII is as high as 10.5 Bohr magneton. Then, the observed behavior looks obvious. What is the author's comment on this? What if authors use other rare earth metals?
There are several questions. The question makes sense in its location. I have attached the manuscript with annotations to make revision easy. Please consider these questions in your revision.

Comments on the Quality of English Language
Minor English language editing is required.
Author Response
Preparing a weakly dimerized stack of tetraselenatetracene (TSeT) and single ion magnetism of the [Dy(III)(hfac)4]- anions combining conducting and magnetic sublattices, Flakina et al. studied the behavior of crystalline {TSeT1.5}.+[DyIII(hfac)4]- salt. The authors have thoroughly investigated the system, and the manuscript is written well. I have a few questions for the authors.
Two broad questions are:
(1) How does conducting behavior and magnetism of {TSeT1.5}.+[Dy(III)(fac)4]- compare with pure TSeT monomers and Dy?
Response: Pure TSeT monomers cannot provide conductivity since conductivity appears in the system only when the triplet state of the dimers is populated. However, the magnetic properties of this salt at low temperatures are mainly defined by TSeT monomers and Dy ion (dimers are diamagnetic below 100 K and does not participate in the magnetic coupling at low temperatures).
(2) How localized is the magnetism resulting from Dy?
Response: Since Dy ions are surrounded by fluoro-substituents of hfac they have very weak coupling with surrounding paramagnetic species. Nevertheless, a slight increase of the XMT values below 50 K can be explained by weak magnetic coupling through the TSeT monomers preserved paramagnetic at low temperatures. This possibility is discussed in the main text.
(3) The magnetic moment of a DyIII is as high as 10.5 Bohr magneton. Then, the observed behavior looks obvious. What is the author's comment on this? What if authors use other rare earth metals?
Response: The observed value of 14.34 emu×K/mol at 300 K corresponds to Myueff = 10.7 Bohr magneton, the same value as shown by the reviewer. The moment is slightly large due to an additional weak contribution from paramagnetic TSeT. We synthesize also other salts with La(hfac)4-, they have similar stricture and also show semiconducting behavior, but singleion magnetism is not found for these salts, such magnetism is observed mainly for Dy(hfac)4-.
There are several questions. The question makes sense in its location. I have attached the manuscript with annotations to make revision easy. Please consider these questions in your revision.
Page 1 ?
Response: Corrected
Page 5 .;
Response: Corrected.
Page 5. This equation appeared out of no where!!!
Response: The equation was shifted to the shown place.
Page 6. What are these components?
Response: The word “components” means Lorentzian components used for fitting of EPR signal.
Where is the Fermi level (in the manuscript).
Response: To show where is the Fermi level we insert a new sentence to the text: “For ease of comparison, all energy levels of trimer and dimer are shifted by the Fermi energy level of the stack (-4.3789 eV).”
Page 10.
What do you mean by full article?
Response: This sentence was corrected: “The results on other crystalline salts of the M(hfac)x- anions (M = metal) with the donors TSeT and BEDT-TTF will be published in our future article.”

Round 2
Reviewer 3 Report
Comments and Suggestions for Authors
The authors adequately addressed my concerns in the earlier version of the manuscript.